# An Eco-Friendly Approach Utilizing Green Synthesized Titanium Dioxide Nanoparticles for Leather Conservation against a Fungal Strain, *Penicillium expansum* AL1, Involved in the Biodeterioration of a Historical Manuscript

**DOI:** 10.3390/biology12071025

**Published:** 2023-07-20

**Authors:** Amr Fouda, Mahmoud Abdel-Nasser, Ahmed M. Eid, Saad El-Din Hassan, Aya Abdel-Nasser, Nada K. Alharbi, Ahlam H. AlRokban, Gomaa Abdel-Maksoud

**Affiliations:** 1Department of Botany and Microbiology, Faculty of Science, Al-Azhar University, Nasr City, Cairo 11884, Egyptsaad_hassan@azhar.edu.eg (S.E.-D.H.); 2Department of Manuscripts Conservation, Al-Azhar Al-Sharif Library, Cairo 11511, Egypt; 3Food Toxicology and Contaminants Department, National Research Centre, Cairo 12622, Egypt; 4Department of Biology, College of Science, Princess Nourah bint Abdulrahman University, Riyadh 11671, Saudi Arabia; nkalharbi@pnu.edu.sa (N.K.A.); ahalrokban@pnu.edu.sa (A.H.A.); 5Conservation Department, Faculty of Archaeology, Cairo University, Giza 12613, Egypt

**Keywords:** green synthesis, titanium dioxide nanoparticles, probiotic bacteria, fungi, biodeterioration, biological control

## Abstract

**Simple Summary:**

The main challenge in libraries, archives, and museums is the fungal deterioration of historical manuscripts (paper and leather), and an eco-friendly approach can be used to reduce or stop biodeterioration. Therefore, the current study examines the green synthesis of titanium oxide nanoparticles (TiO_2_-NPs) using a green method to inhibit the growth of fungal strains isolated from deteriorated historical manuscripts. Fifteen fungal strains were involved in the deterioration of historical manuscripts dating back to the 16th century that were identified by traditional and molecular identification. In addition, their potential in biodeterioration was investigated by hydrolytic enzymatic activities. TiO_2_-NPs were synthesized using a probiotic bacterial extract as a safe method and characterized by UV-Vis spectroscopy, X-ray diffraction (XRD), transmission electron microscopy (TEM), energy dispersive X-ray (EDX), and Fourier-transform infrared (FT-IR). The safe concentration of synthesized TiO_2_-NPs that was used as a coating agent for leather to inhibit fungal growth was detected via the investigation of in vitro cytotoxicity against two normal cell lines (skin and lung). The experimental study showed that the leather without NP treatment and inoculated with the most potent fungal strain exhibited high deterioration compared with those coated with NPs and inoculated with fungal strain.

**Abstract:**

The main hypothesis of the present research is investigating the efficacy of titanium oxide nanoparticles (TiO_2_-NPs) to prevent the growth of fungal strains when applied on leather under an experimental study. Therefore, fifteen fungal strains were isolated from a deteriorated historical manuscript (papers and leathers) and identified by traditional methods and ITS sequence analysis, including *Aspergillus chevalieri* (one isolate), *A. nidulans* (two strains), *A. flavus* (four strains), *A. cristatus* (one strain), *A. niger* (one strain), *Paecilomyces fulvus* (two strains), *Penicillium expansum* (two strains), and *P. citrinum* (two strains). The enzymes cellulase, amylase, pectinase, and gelatinase, which play a crucial role in biodegradation, were highly active in these fungal strains. TiO_2_-NPs were formed using the cell-free filtrate of the probiotic bacterial strain, *Lactobacillus plantarum*, and characterized. Data showed that the TiO_2_-NPs were successfully formed with a spherical shape and anatase phase with sizes of 2–8 nm. Moreover, the EDX analysis revealed that the Ti and O ions occupied the main component with weight percentages of 41.66 and 31.76%, respectively. The in vitro cytotoxicity of TiO_2_-NPs toward two normal cell lines, WI38 and HFB4, showed a low toxicity effect against normal cells (IC_50_ = 114.1 ± 8.1µg mL^−1^ for Wi38, and 237.5 ± 3.5µg mL^−1^ for HFB4). Therefore, concentrations of 100 μg mL^−1^ were used to load on prepared leather samples before inoculation with fungal strain *P. expansum* AL1. The experimental study revealed that the loaded TiO_2_-NPs have the efficacy to inhibit fungal growth with percentages of 73.2 ± 2.5%, 84.2 ± 1.8%, and 88.8 ± 0.6% after 7, 14, and 21 days, respectively. Also, the analyses including SEM, FTIR-ART, color change, and mechanical properties for leather inoculated with fungal strain AL1 in the absence of NPs showed high damage aspects compared to those inoculated with fungal strains in the presence of TiO_2_-NPs.

## 1. Introduction

The significant advancements in metal and metal oxide nanoparticles (NPs) have paved the way for their incorporation in a wide range of biomedical and biotechnological applications, offering an appealing alternative source for such purposes [1]. Recently, ZnO-NPs, CuO-NPs, Fe_2_O_3_-NPs, and MgO-NPs have been extensively used in various biological applications due to their unique physiochemical characteristics [2,3]. Among metal oxide NPs, TiO_2_ has attracted more attention from researchers due to its unique properties such as chemical and thermal stability, high surface area, nontoxicity, surface morphology, biocompatibility, and exceptional dielectrically and optical properties [4]. Therefore, it was used in various applications such as solar cells, optical, electronics, sensors, gas sensing, magnetics, lithium batteries, photocatalysis, H_2_O_2_ reduction, optoelectronic devices, sunscreens, paints, pigments, antimicrobials, antifungal agents, drugs, skin care products to protect skin from harmful UV-rays, food colorants, plastics, toothpaste, inks, and papers [4,5,6,7]. It has been found that TiO_2_-NPs are efficient against a variety of microbes, and their antimicrobial action is broader than any other biocide, particularly in their nanosized form [8]. 

Researchers have recently focused on developing quick, efficient, safe, and affordable treatment methods to protect heritage culture assets from microbial harm. Due to the rapid advancement of nanotechnology science and its efficiency against a variety of microorganisms, researchers have utilized NPs, particularly those produced using environmentally friendly methods, in the preservation processes [9]. The biological or green synthesis of NPs utilizing metabolites of various biological entities (bacteria, fungi, actinomycetes, yeasts, and plants) has received more consideration than chemical and physical strategies due to being safe, eco-friendly, cost-effective, avoiding harsh conditions, easy scale-up, and biocompatible strategy [3,10]. Since *Lactobacillus* spp. is a common cause of milk curdling, it is advantageous because it is non-pathogenic, oxygen resistant, and has high metabolic fluxes. Due to the large number of metabolites produced by *Lactobacillus*, it is a helpful microorganism used for the environmentally friendly synthesis of highly stable nanomaterials [11].

Many historical and archaeological manuscripts that can be found in libraries, archives, museums, and storage facilities constitute invaluable national treasures for all people on the planet. Most signs of degradation, including stains, erosion, and weakness, can be observed on historical manuscripts and may be caused by unsuitable physical (light, temperature, and relative humidity), chemical (pollutants and particulates), and biological (insects and microbes) factors [12,13]. One of the most significant causes of degradation in library and archive collections is the biodeterioration of bookbinding made of vegetable-tanned leather by fungi, which are among the most harmful species [14,15]. The materials are deeply penetrated by the fungal hyphal networks, which causes significant loss from acid corrosion, enzymatic degradation, and mechanical attack [16,17]. The most common fungi that were associated with the deterioration manuscript and contributed to the biodeteriorations, according to previously published research, were *Aspergillus niger*, *A. flavus*, *A. terreus*, *A. ochraceous*, *A. carbonarius*, *A. fischeri*, *A. fumigatus*, *A. tamarii*, *Eurotium chevalieri*, *Cladosporium cladosporioides*, *Fusarium poae*, *Wallemia sebi*, *Penicillium notatum*, *P. oxalicum*, *P. rubrum*, and *Aletrnaria alternate* [18,19]. 

Therefore, the main hypothesis of the current investigation is to determine whether environmentally friendly TiO_2_-NPs fabricated by cell-free filtrate of *Lactobacillus plantarum* DSM 20174 are effective to control the highly deteriorated fungal strains when inoculated on the prepared leather samples. To achieve this hypothesis, a degraded historical manuscript (paper and leather) was chosen as a source for fungal isolation, which identified both traditional and molecular techniques. Through an analysis of their enzymatic activity, isolated fungal strains were found to play a part in biodegradation. Utilizing UV-Vis spectroscopy, X-ray diffraction (XRD), transmission electron microscopy (TEM), energy dispersive X-ray (EDX), and Fourier-transform infrared (FT-IR), the TiO_2_-NPs were characterized. Finally, a safe dose of TiO_2_-NPs, which were detected by testing their cytotoxicity against normal lung and skin cell lines, has been employed to inhibit fungal growth. An experimental study including inoculation of the fungal strain on goatskin leather after being treated with TiO_2_-NPs was investigated. The fungal growth inhibition percentages were calculated after 7, 14, and 21 days. Moreover, various techniques such as environmental scanning electron microscope (ESEM), attenuated total reflectance Fourier-transform infrared (FTIR-ART) spectroscopy, measurements of color change, and mechanical properties have been used to evaluate the physicochemical properties of leather treated with NPs in the presence and absence of fungal inoculation.

## 2. Materials and Methods

### 2.1. Historical Manuscript Studied

The historical manuscript named “Tabsir Al-Rahman Wa-Taysir Al-Manan Baed Ma Yushir AlaaAaejaz Alquran” dated back to the 16th century and is preserved in the basement of the Bibliotheca Alexandrina Library, Egypt. This manuscript was written in the Arabic Language, contains one volume with a size (cm^3^) of 25 length × 17 width × 5 height, and is composed of 344 sheets collected by leather binding. This manuscript is one of the collections donated to the library.

The environmental conditions when sampling were as follows: temperature 20 °C, relative humidity was high, and the manuscript was preserved on iron shelves. The storage cabinets in the basement are below the surface of the earth and directly adjacent to the Mediterranean; thus, the ventilation is poor, and the relative humidity is high.

### 2.2. Visual Assessment to Detect the Biodeterioration Aspects

A digital camera (Samsung 38 MP, f/2.2 lens slot, Tokyo, Japan) and the human eye were both utilized to observe the deterioration signs on the surface of the historical manuscript [20].

### 2.3. Isolation and Identification of Fungal Strains Associated with Historical Manuscripts

For fungal isolation, sterilized cotton swabs were used and pressed over the deteriorated area before being transferred to the lab to serve as a source for isolation. Each swab was placed into a test tube containing 10 mL of sterile saline solution (to enhance or reactivate the fungal spore) supplemented with 500 mg L^−1^ chloramphenicol (as an antibacterial agent) for 24 h. After that, 50 µL of the previous saline solution was spread using a sterilized glass rod on the surface of the Czapeck Yeast Extract (CYA) agar plate and Potato Dextrose Agar (PDA) plate, which was incubated at 25 °C for 5 days and observed daily. Once the fungal colony appeared, it was reinoculated into a new plate for forthcoming purification. Each purified fungal isolate was preserved on the CYA and/or PDA slant at 4 °C for further investigation.

All acquired fungal isolates were identified based on cultural, morphological, and microscopic inspection using standard keys for *Aspergillus* spp. [21], *Penicillium* spp. [22], *Eurotium* spp. [23,24], and *Paecilomyces* spp. [25]. 

After that, ITS sequence analysis was used to molecularly identify the isolated fungal strains. Eight fungal isolates (one of each strain based on conventional identification) were chosen for ITS identification. Fungal DNA was extracted using the Gene Jet Plant genomic DNA purification kit (Thermo Fisher Scientific Inc., Waltham, MA, USA). Fungal DNA served as the primer set for the PCR that amplified the ITS region. For ITS1 and ITS4, the primers 5-TCCGTAGGTGAACCTGCGG-3 and 5-TCCTCCGCTTATTGATATGC-3, respectively, were used [26]. Using the Maxima Hot Start PCR Master Mix (Thermo), 0.5 µM of each primer and 1 µL of isolated fungal DNA were added to 50 µL of the PCR tube. The PCR protocol was completed according to Fouda et al. [9] using a DNA Engine Thermal Cycler from the Sigma Scientific Services Company (located in Cairo, Egypt). The PCR reaction involved a hot start at 94 °C for 3 min, followed by 30 cycles consisting of 94 °C for 30 s, 55 °C for 30 s, and 72 °C for 1 min. A final extension step was performed at 72 °C for 10 min. The sequencing process was carried out at the GATC Company (Konstanz, Germany) using an ABI 3730 × 1 DNA sequencer. Using an NCBI-BLAST tool, the resulting sequences were compared to ITS sequences that were deposited in GenBank. The phylogenetic tree was made using the neighbor joining method and confidence-tested bootstrap analysis (MEGA v6.1, www.megasoftware.net, accessed on 19 January 2013).

### 2.4. Enzyme Activity

The ability of the fungal strains isolated from damaged ancient manuscripts to release various enzymes, such as cellulase, amylase, gelatinase, and pectinase, was assessed using the agar plate method. Each fungal isolate was inoculated on a mineral salt agar (MSA, containing g L^−1^: NaNO_3_, 5; KH_2_PO_4_, 1; K_2_HPO_4_, 2; MgSO_4_·7H_2_O, 0.5; KCl, 0.1; CaCl_2_, 0.01; and FeSO_4_·7H_2_O, 0.02; agar, 15; dis. H_2_O, 1 L) plate supplemented with 1% of a specific substrate (carboxymethyl cellulose for cellulase enzyme, starch for amylase, gelatin for gelatinase, and pectin for pectinase), which was inoculated with a new disc (0.5 mm in diameter) in the center. Chloramphenicol was added to MSA media as an antibacterial agent to inhibit bacterial growth. The inoculation plates underwent a 96 h incubation period at 25 °C before being flooded with a particular reagent. The results were calculated as follows: diameter of the entire clear zone − diameter of the fungal growth.

Gelatinase activity was evaluated after the plate had been inundated with acidic mercuric chloride, whereas the cellulase, amylase, and pectinase activities were investigated after the plate had been flooded with an iodine solution [27]. 

### 2.5. Biosynthesis of Titanium Oxide Nanoparticles (TiO_2_-NPs)

#### 2.5.1. Probiotic Bacterial Strain

*Lactobacillus plantarum* DSM 20174 as a probiotic strain was used to evaluate the potential of their metabolites to fabricate TiO_2_-NPs. This strain was bought from the Faculty of Agriculture, Ain Shams University, Cairo, Egypt’s Microbiological Resources Centre (MIRCEN). 

#### 2.5.2. Preparation of Cell-Free Filtrate and Biosynthesis of TiO_2_-NPs

The MRS broth media (Ready-prepared, Merck, Darmstadt, Germany) was inoculated with a single colony of *L. plantarum* and incubated at 35 ± 2 °C for 24 h. After that, the inoculated broth media were centrifuged at 1000 rpm for 10 min to obtain the cells, which were collected and rinsed twice with sterile deionized water before being resuspended (10 g) in 100 mL of deionized water and incubated for 24 h at 35 ± 2 °C. The mixture was centrifuged after the incubation period to obtain the supernatant (cell-free filtrate, CFF), which was then utilized as a biocatalyst for the biosynthesis of TiO_2_-NPs by mixing with titanium isopropoxide (Ti[OCH (CH_3_)_2_]_4_) under stirring conditions (for 1 h) to obtain a final concentration of 5 mM [28]. For 1 M NaOH, which was added drop by drop, the mixture’s pH was adjusted to 8. The appearance of a white precipitate indicates the successful formation of TiO_2_-NPs, which were collected after 24 h of incubation, washed three times with ultrapure water (Milli-Q), and then dried in an oven for 3 h at 200 °C.

#### 2.5.3. Characterization of TiO_2_-NPs

The first sign of the synthesis of TiO_2_-NPs is the color shift of the CFF from pale yellow to white after mixing with the metal precursor. The absorbance of the white color was then determined by measuring their absorbance using UV-visible spectroscopy (JENWAY 6305, Staffordshire, UK) at wavelengths between 200 and 800 nm. To identify the highest surface plasmon resonance (SPR), 2 mL of the synthesized solution was placed in a quartz cuvette, and its absorbance was then monitored at regular intervals wavelengths [28]. 

Using Fourier-transform infrared analysis, the functional groups in the bacterial CFF were compared to TiO_2_-NPs (FT-IR, Cary-660 model). This procedure involved mixing 10 mg of synthetic TiO_2_-NPs or 5 mL of cell-free filtrate with KBr thoroughly, pressing the mixture under pressure to create a disc, and then scanning it at wavenumbers between 400 and 4000 cm^−1^ [29].

X-ray diffraction (XRD, PANalytical-X’Pert-Pro-MRD) was used to examine the natural structure (crystallinity or amorphous) of synthetic TiO_2_-NPs. The operating conditions were achieved at a current and voltage of 30 mA and 40 Kv, respectively, using CuK (λ = 1.54) as the X-ray source. The XRD scanning was performed at 2θ values between 5 and 80°. According to the Debye equation, Scherrer’s typically sized crystallites are biosynthesized TiO_2_-NPs, as determined by XRD examination [30]: (1)$$\mathrm{Average}\;\mathrm{crystallite}\;\mathrm{size}=\frac{0.94\times 1.54}{\beta \cos\theta }$$
where 0.94 is the Scherrer constant, 1.54 is the wavelength of the X-ray, β is the full width of the diffraction peak at a half maximum, and θ is the diffraction angle.

The sizes, shapes, surface morphology, and chemical compositions of biosynthesized TiO_2_-NPs were discovered by using a transmission electron microscope (TEM, JEOL, Ltd-1010, Tokyo, Japan) and a scanning electron microscope (SEM, JEOL, JSM-6360LA, Tokyo, Japan) connected with energy dispersive X-ray (EDX). The as-formed powder was suspended in high pure water (Milli-Q) under sonification followed by drooping a few drops on the surface of the TEM carbon grid. The loaded grid remains dry before being subjected to analysis. The elementary qualitative and quantitative mapping was detected by EDX connected with the SEM apparatus. 

#### 2.5.4. In Vitro Cytotoxicity of TiO_2_-NPs

To determine a safe dose of TiO_2_-NPs that can be used as a treatment for leather to inhibit the growth of fungi after inoculated on it, the cytotoxic effect against two normal cell lines, WI38 (human fibroblast lung tissue) and HFB4 (human normal melanocytes), was investigated using the MTT (3-(4,5-dimethylthiazol-2-yl)-2,5-diphenyl tetrazolium bromide) method. The chosen normal cell lines were acquired from VACSERA, a holding company for biological products and vaccines, in Cairo, Egypt. With a cell density of 1 × 10^5^ cells/100 µL/well, each kind of cell was separately added to a 96-well culture plate before being incubated for 24 h at 37 °C with 5% CO_2_. The monolayer sheet was created after the incubation period and mixed with 100 µL of a maintenance medium (RPIM, MERCK, Sigma Aldrich, Cairo, Egypt) with 2% serum. The TiO_2_-NPs concentration was doubled (1000, 500, 250, 125, 62.5, and 31.25 μg mL^−1^) and applied to the developing cells before incubation for 48 h. Three untreated wells (containing growing cells in the absence of TiO_2_-NPs) served as a control group. After the incubation period, the remaining growth media in each well was removed and replaced with 50 µL of an MTT solution (5 mg/mL phosphate buffer saline solution), which was then incubated for 4 h at 37 °C. The formazan crystal that had developed as a result of the MTT assay metabolism was then dissolved by adding 100 µL of 10% DMSO to the well after the MTT solution was discarded. The DMSO was removed from the wells after 30 min, and an ELIZA reader was used to measure the color absorbance at 570 nm. [31]. The percentages (%) of cell viability due to TiO_2_-NP treatment were measured using the following equation:(2)$$\mathrm{Cell}\;\mathrm{viability}\;(\%)=\frac{\mathrm{Absorbance}\;\mathrm{of}\;\mathrm{treatment}}{\mathrm{Absorbance}\;\mathrm{of}\;\mathrm{control}}\times 100$$

### 2.6. Experimental Study

#### 2.6.1. Fungal Used

An experimental study was performed to explore the physicochemical properties of leather treated with a safe dose of TiO_2_-NPs (100 µg mL^−1^, which was chosen based on the in vitro cytotoxicity test) in the presence and absence of fungal growth. To achieve this goal, the fungal strain *Penicillium expansum* AL1 was selected as the most enzymatic producer between the fungal strains isolated from historical leather. 

#### 2.6.2. Preparation of Leather Samples Used

The vegetable-tanned (mimosa) goatskin samples were prepared by the authors according to the standard [32]. The prepared leather samples were used with an average diameter of 12.5 cm and a thickness of 0.166 mm without any additives to reduce the number of variables that can influence the outcomes.

#### 2.6.3. Experimental Design

The leather was sterilized by immersing it in 70% ethyl alcohol for 10 min, drying it for 2 h at 37 ± 2 °C, submerging it for two minutes in a solution containing TiO_2_-NPs (100 µg mL^−1^), and then allowing it to dry for an hour inside a vertical laminar flow chamber. A Czapex Dox agar was prepared, sterilized, and poured into sterile glass Petri dishes (diameter = 15 cm). After solidification, the treated leather samples were added to the Petri plate surface. Three replicate samples were performed for each treatment and inoculated with three discs (5 mm) of newly cultivated fungal strain AL1 before being incubated at 25 ± 2 °C for 21 days [33]. The disc was prepared using a sterilized cork borer (5 mm), and each disc was covered by heavy aerial mycelia. The following treatment was conducted: (A) negative control using sterilized leather without either treatment or inoculation, (B) positive control with untreated sterilized leather inoculated by fungal strain AL1, (C) sterilized leather treated with TiO_2_-NPs without fungal inoculation, and (D) sterilized leather previously treated with TiO_2_-NPs and inoculated with fungal strain AL1. The following parameters were evaluated after different time intervals. 

A.Assessment of Fungal Growth

After 7, 14, and 21 days of incubation, the colony measurement method was used to measure fungus growth. Due to its greater sensitivity compared to dry weight, hyphal length, and ergosterol quantification, the colony measurement method (mm) was chosen [34]. The growth inhibition percentage (%) due to TiO_2_-NP treatment was calculated according to the following formula.
(3)$$\mathrm{Inhibition}\;\mathrm{percentages}\;(\%)=\frac{D_{c}-D_{t}}{D_{c}}\times 100$$
where D_c_ = the average of fungal growth in the control (mm) and D_t_ = the average of fungal growth in the treatment (mm).

B.Environmental Scanning Electron Microscope (ESEM)

Using a scanning electron microscope (SEM) (Quanta 3D 200i produced by FEI, accelerated voltage of 20.00 kV, and a magnification range of 250 to 2000×), the surface morphology of leather after 21 days in the presence and absence of fungal development was investigated. The Great Egyptian Museum—Conservation Centre (GEM.CC) in Cairo, Egypt, conducted the SEM investigation [35].

C.Attenuated Total Reflectance Fourier-Transform Infrared (FTIR-ART)

FT-IR analysis was performed to compare different treatments of leather samples after 21 days with the control [36]. FTIR-ART spectra on a Bruker Vertex 70 Platinum ATR scale with crystal diamonds in the range of 4000–400 cm^−1^ and a resolution of 4 were utilized by the Archeological Research and Preservation Centre—Supreme Council of Antiquities—Ministry of Tourism and Antiquities, Egypt, to analyze the leather samples.

D.Measurement of Color Change

After 7, 14, and 21 days, color change and total color difference values of the leather samples were assessed using the CIELAB technique (portable spectrophotometer by Hunter Lab-Reston, VA, USA). The CIE system consists of two channels, one of which measures the change from red to green (a*) and the other measures the change from yellow to blue (b*). It also has a channel for measuring lightness, abbreviated L*. The following equation (∆E) was used to calculate the overall color difference [9].
(4)$$\Delta E=\sqrt{{\left(\Delta L\right)}^{2}+{\left(\Delta a\right)}^{2}+{\left(\Delta b\right)}^{2}}$$

E.Mechanical Properties Measurement

The mechanical properties (tensile strength and elongation) of the treated samples before and after the inoculation were studied using the dynamometer produced by SDL ATLAS, H5KT at the National Institute for Standards (NRC), Haram, Giza, Egypt. Samples were taken from adjacent zones of skin. These samples were 90 mm long, the free length between the jaws of the testing machine was 50 mm, and the width of the free length was 10 mm. The elongation at break is given in %. Five measurements of each sample were used to calculate the average tensile strength and elongation. It should be highlighted that both before and after infection, all measurements were made in comparison to the control sample [37]. 

### 2.7. Statistical Analysis

The data in the current study are represented by the means of three independent replicates using the statistical application SPSS v17 and are then subjected to an ANOVA analysis. The Turkey HSD test was used to examine the mean difference comparison between the treatments; it had a 0.05 *p*-value.

## 3. Results and Discussion

### 3.1. Investigate the Biodeterioration Aspects of Historical Manuscript Using Visual Observation before Being Used as a Source for Fungal Isolation

Photographic documentation using a digital camera (Figure 1) clearly showed signs of deterioration in the manuscript’s paper and leather binding. The three degradation-related traits that were most frequently seen were hardness, rigidity, and lack of flexibility. Vichi et al. [38] claim that the types of deterioration mentioned above may be due to the alteration and degradation of the heterogeneous composition of leathers by the action of light, humidity, temperature, and pollutants.

The degradation of leather may finally result in the material’s disintegration and harm to the document. According to Sebestyén and co-authors, there are internal and external variables that contribute to different elements of leather degeneration [39]. Leather can eventually degrade to the point where it disintegrates, damaging the paper. According to Sebestyén, various aspects of leather deterioration are influenced by both internal and extrinsic factors. The internal features are a product of how leather goods are created. In addition to biological (insects and microorganisms) and environmental (light exposure to radiation, humidity levels, temperature, biochemical and particle matter pollution, and naturally occurring occurrences, especially those connected to climate change) factors, there are also external influences. Moreover, certain stains that may be the result of a microbial infection were seen [39].

Historical paper manuscripts (Figure 1C–E) showed certain signs of deterioration, such as yellowing, paper fragility, and weakening. Paper deterioration types may be attained due to a rise in acidity, which may be brought on by an external or internal component [36,40]. When cellulose ages, it naturally creates acids such as formic, acetic, lactic, and oxalic acids (pollutants from surrounding environmental conditions). Because of the potent intermolecular interactions, the paper does not easily release these acids. The presence of moisture from permanent or external sources caused the glucose chains to continuously break down into shorter lengths. More acids are produced as a result of the hydrolysis reaction, which promotes further degradation. Water stains and the paper’s ink’s distortion were other things that could be seen. This kind of degeneration showed how the document under examination was harmed by prolonged exposure to high relative humidity. The stains on the historical papers might have been caused by the growth of bacteria and fungi, and have a significant role in biodeterioration [9].

### 3.2. Fungal Isolation and Identification

In the current study, the different fungal strains that inhabit the deteriorated parts of leather binding and historical papers were isolated. As shown, fourteen fungal strains were obtained, including four isolates (AL1, AL3, AL5, and AL13) from historical leather bookbinding and the remaining eleven from deteriorated historical papers. All fungal strains have undergone morphological, cultural, and microscopic identification. The results reveal that the fungal isolates AL1 and AL13 were identified as *Penicillium expansum*, whereas the fungal isolates AL2 and AL10 were identified as *Eurotium* spp. On the other hand, the fungal isolates AL3 and AL8 were identified as *Aspergillus nidulans*, AL4, AL5, AL9, and AL12 were identified as *A. flavus*, and one isolate (AL11) was designated as *A. niger*. Finally, the remaining four fungal isolates (AL16 and AL17) and (AL14 and AL15) were identified as *Paecilomyces* spp. And *Penicillium citrinum,* respectively (Table 1). 

According to the identification data, the most common fungal strains associated with historical manuscripts were arranged as follows: *Aspergillus* spp. followed by *Penicillium* spp., *Eurotium* spp., and *Paecilomyces* spp. with percentages of 46.7, 26.7, 13.3, and 13.3%, respectively. Among *Aspergillus* spp., the most common strains were *A. flavus* with 57% followed by *A. nidulans* and *A. niger* with percentages of 28.6% and 14.3%, respectively. On the other hand, *P. expansum* and *P. citrinum* were equal in their percentages. In a similar study, 20 fungal strains isolated from deteriorated papers of European origin were identified using cultural and microscopic examination as *Aspergillus* spp., *Penicillium* spp., *Eurotium* spp., and mycelia sterile hyphae (their structures are not distinguished) with percentages of 45, 35, 5, and 15%, respectively [17]. Also, traditional identification methods were used to identify the fungal strains associated with historical leather bookbinding and papers [36]. The authors reported that the most common fungal strains isolated from historical papers were *A. flavus* and *A. niger*, whereas the fungal strains *A. terreus*, *A. niger*, and *A. flavus* were the most common strains isolated from leather bookbinding. Mansour et al. reported that the fungal strains *Cladosporium cladosporioides*, *Eurotium chevalieri*, *Fusarium poae*, *Wallemia sebi*, *A. fumigatus*, and *A. tamarii* were the most common fungi isolated from leather bookbinding dating back to 18th century [41]. 

The traditional identification was confirmed using molecular methods through amplification and sequencing of the ITS gene. The fungal strains coded AL1, AL2, AL3, AL4, AL6, AL10, AL11, and AL14 were selected to confirm their identification. These strains were chosen based on one strain from each species. The gene sequence analysis reveals that the selected fungal isolates were like *Penicillium expansum*, *Aspergillus chevalieri*, *A. nidulans*, *A. flavus*, *Paecilomyces fulvus*, *A. cristatus*, *A. niger*, and *Penicillium citrinum* with similarity percentages of 99.14, 99.56, 99.48, 98.78, 99.24, 99.03, 99.06, and 97.98%, respectively. The closest accession number for each strain was recorded in Table 1. Hence, the selected fungal isolates were identified as *P. expansum* AL1, *A. chevalieri* AL2, *A. nidulans* AL3, *A. flavus* AL4, *Paecilomyces fulvus* AL6, *A. cristatus* AL10, *A. niger* AL11, and *P. citrinum* AL14 (Figure 2, Table 1). The sequences of the selected fungal strains were deposited in GenBank and given accession numbers, as shown in Table 1.

The fungal strains isolated in the current study are involved in biodeterioration and material spoilage, as mentioned previously. They can be found in leathers, papers, building materials, softwood, dust, and textiles [42,43]. Also, the isolated fungal strains, such as *Aspergillus* and *Eurotium,* are xerophytic molds that are considered one of the most indoor common fungal communities in libraries, museums, and archeological buildings, and play a critical role in biodeterioration [44]. These activities could be related to their efficacy to secrete a wide range of hydrolytic enzymes and various metabolites, such as acids [16,45]. Interestingly, the purpose of isolating and identifying the diverse fungal communities present in libraries, museums, and archives extends beyond investigating their role in biodeterioration. It also encompasses addressing the potential risks they pose to the health of workers and visitors. Different strains belonging to *Aspergillus*, *Penicillium*, *Rhizopus*, *Cladosporium*, *Paecilomyces*, and *Eurotium* were isolated from libraries, archives, and museums and their health problems such as allergic infections, cutaneous inflammation, aspergillosis, otomycosis, respiratory infections, and endocarditis were investigated [18,46].

### 3.3. Enzymatic Activities

As fungi grow in the fibers of historical manuscripts and create hydrolytic enzymes, including cellulase, amylase, xylanase, and pectinase, as well as acids or pigments that contribute to degradations, they are believed to be the primary degrading agents for historical manuscripts. The paper industry uses proteins, polysaccharides, gelatin, starch flour, and various synthetic substances in addition to organic components (mostly cellulose) to reduce ink spread and promote fiber connection [47]. Fungi can degrade and change the molecular makeup of these materials by dissolving organic compounds and other additives via extracellular hydrolytic enzymes [48]. Several hydrolytic enzymes, including cellulase, amylase, pectinase, and gelatinase, can be secreted by fungal strains and showed a significant role in biodeterioration. According to the obtained data, fifteen fungal strains can emit varying amounts of cellulase, amylase, pectinase, and gelatinase (Figure 3). Data analysis showed that the highest cellulase activity was recorded for fungal strain AL15 with an inhibition zone of 18.7 ± 0.6 mm followed by fungal strain AL13 with inhibition zones of 12.3 ± 2.3 mm, while the fungal strain AL14 did not exhibit any cellulase activity (Figure 3). Interestingly, the cellulase activity between fungal strains AL1 and AL12 was not significant (*p ˂* 0.001), with inhibition zones ranging between 3.3 ± 1.5 mm and 5.7 ± 1.5 mm. In a similar study, *P. chrysogenum* and *A. niger*, which were isolated from a historical book from the 17th and 18th centuries, were chosen as the highest cellulase producers [9]. Following the colonization of paper by fungal colonies, hyphae penetrated the fibers and secreted the cellulase enzyme, causing damage to the glycosidic linkage between fibers in addition to some physical changes [49].

Analysis of variance showed that the highest amylase activity was recorded for fungal strains AL13 and AL1, with clear zones of 11.7 ± 3.8 mm and 8.0 ± 1.7 mm, respectively, followed by fungal strains AL6 and AL3 with clear zones of 7.0 ± 1.0 mm and 6.0 ± 1.7 mm, respectively (Figure 3). Interestingly, the amylase activity between fungal strains AL1–AL4, AL6, and AL10–AL12 was not significant (*p ˂* 0.001), with inhibition zones ranging between 2.7 ± 1.2 mm and 8.0 ± 1.73 mm. Moreover, all fungal strains have the activity to secrete gelatinase enzymes with the highest clear zones of 6.7 ± 1.5 mm and 5.7 ± 2.3 mm for fungal strains AL1 and AL12, respectively, while the fungal strains AL11 and AL14 do not exhibit any gelatinase activity (Figure 3). Interestingly, the gelatinase activity between fungal strains AL4 and AL10 was not significant (*p ˂* 0.001). Unfortunately, fungal strain AL14 did not exhibit pectinase activity, whereas fungal isolates AL7 and AL8 were the highest pectinase producers, with clear zones of 6.7 ± 1.5 mm and 5.7 ± 2.1 mm, respectively (Figure 3).

Macromolecules can be split into smaller pieces by hydrolytic enzymes like cellulase and amylase, which convert cellulose and starch into glucose monomers [45]. The enzymes pectinase and gelatinase can degrade fibroin, collagen, and keratin, which are used to make silk, wool, and parchment, respectively [50]. Accordingly, the effectiveness of these fungal strains in secreting several enzymes, including amylase, pectinase, cellulase, and gelatinase, with varied degrees, determines their role in the biodeterioration process. The current study’s storage environment, which had a high RH and temperature limit of 25 °C, is regarded to be the perfect habitat for fungi to thrive and, as a result, release a variety of active metabolites, including enzymes and acids. The acidic secretions are what lead to the acid hydrolysis of historical materials [43]. 

In addition to their effectiveness in producing a wide range of enzymes, the isolated fungal strains can be distinguished by their capacity to release acidic metabolites. The fungal strains *A. terreus*, *A. niger*, *A. versicolor*, *A. ustus*, *Cladosporium* sp., *Penicillium commune*, *P. chrysogenum*, *P. expansum*, and *P. citrinum* can emit acidic metabolites that can lower pH levels up to 4.0 [13]. 

### 3.4. Biosynthesis of Titanium Dioxide Nanoparticles (TiO_2_-NPs) Using Lactobacillus plantarum

#### 3.4.1. UV-Vis Spectroscopy

The synthesis of TiO_2_-NPs using biological approaches, such as microbial fermentation or the use of microbial byproducts, has garnered interest due to its eco-friendly nature and potential applications [51]. Various microorganisms, including bacteria and fungi, have been explored for the synthesis of TiO_2_ nanoparticles [52]. UV-visible spectroscopy proved to be an effective method for determining the complete reduction in precursors. The color change was visibly observed in the cell-free filtrate (CFF) of *L. plantarum* during the incubation with a Ti[OCH(CH_3_)_2_]_4_ solution. In contrast, the pure Ti[OCH(CH_3_)_2_]_4_, without the presence of *L. plantarum* CFF, did not exhibit any discernible color variation. The formation of white color following the mixing of the CFF with a metal precursor indicates the formation of TiO_2_-NPs. The intensity of color was measured at an optical range from 200 to 800 nm. Figure 4A represents the UV-Vis spectroscopy of TiO_2_-NPs that showed the maximum absorption peak at 365 nm, which was a primary sign of the successful formation of TiO_2_-NPs. The obtained result is compatible with published investigations, which state that the surface plasmon resonance for TiO_2_-NPs has been in the ranges of 300–400 nm [53,54].

#### 3.4.2. Fourier-Transform Infrared Spectroscopy (FT-IR)

The provided data describes the FT-IR spectra of the CFF of *L. plantarum* and TiO_2_-NPs synthesized using the same CFF. The FT-IR spectra are used to identify the functional groups present in the samples based on their characteristic absorption peaks [29]. For the CFF of *L. plantarum*, five peaks were observed at the following wavenumbers: 3305, 1956, 1527, 1304, and 1019 cm^−1^. These peaks were shifted to wavenumbers of 3585, 1649, 1087, 617, and 549 cm^−1^ for bacterial synthesized TiO_2_-NPs (Figure 4B). The peak at a wavenumber of 3305 cm^−1^ suggests the presence of O-H stretching vibrations, typically associated with hydroxyl groups, which overlapped with the N–H stretching of aliphatic primary amines [55,56]. This peak was shifted to 3585 after TiO_2_-NP fabrication. The peak at 1956 cm^−1^ corresponds to a high-wavenumber band that may indicate the presence of carbonyl groups (C=O stretching vibrations), such as those found in ketones, aldehydes, or carboxylic acids [57,58]. The aromatic C=C stretching vibrations, which are often present in compounds containing aromatic rings are represented by a peak at 1527 cm^−1^ [8,59], which shifted to 1649 cm^−1^ at TiO_2_-NPs. The peaks at wavenumbers of 1304 and 1019 cm^−1^ could indicate the presence of C-O stretching vibrations, commonly found in alcohols, ethers, or esters [6]. These peaks were shifted to 1087 cm^−1^ after the formation of TiO_2_-NPs. The appearance of new peaks at wavenumbers 617 and 549 cm^−1^ could be attributed to Ti-O stretching vibrations, which are characteristic of TiO_2_-NP compounds [60].

#### 3.4.3. X-ray Diffraction (XRD)

The XRD analysis of TiO_2_-NPs revealed peaks at specific 2θ values. These peaks correspond to the diffraction pattern of the crystalline structure of the nanoparticles. Figure 5A showed the presence of eight diffraction peaks at 2θ° of 25.4°, 37.5°, 47.7°, 53.6°, 54.5°, 62.7°, 70.1°, and 74.7°, which corresponded to plans of (101), (004), (200), (105), (211), (204), (220), and (216), respectively. The anatase phase of biosynthesized TiO_2_-NPs was confirmed by various diffraction peaks obtained following the JCPD standard (No. 2 21-1272). This result agreed with previous reports on the anatase formula of TiO_2_-NPs [5,11]. The observation of a prominent peak in the XRD pattern within the 2θ range of 25.25°–25.58° signifies the presence of the crystallographic plane (101), indicating the successful synthesis of anatase crystalline structure [61]. The absence of additional peaks indicates the purity of synthesized TiO_2_-NPs. By employing the Debye–Scherrer equation, the XRD analysis allowed for the determination of the crystallite size of the synthesized TiO_2_-NPs. Specifically, the calculation was based on the primary crystal plane (101), yielding a crystallite size of 9 nm. Similarly, the crystallite sizes of TiO_2_ fabricated by *Acacia nilotica* and orange peel aqueous extract were calculated using XRD analysis and were found to be 9 and 17.3 nm, respectively [7,8]. 

#### 3.4.4. Transmission Electron Microscopy (TEM) and Energy Dispersive X-ray (EDX)

The provided data pertains to the TEM analysis of TiO_2_-NPs (Figure 5B). The analysis revealed that the particle sizes ranged from 2 to 8 nm, indicating a relatively small size distribution. Additionally, the particles exhibited a spherical shape. The TEM technique is widely used for visualizing and characterizing nanoparticles at high resolution. It allows for the direct observation of individual particles and provides valuable information about their size, morphology, and overall structure [1,62]. The data suggests that the TiO_2_-NPs synthesized in this study possess a relatively narrow size range, with particles ranging from 2 to 8 nm. The presence of nanoparticles within this size range indicates that the synthesis process yielded uniform-sized particles, which can have significant implications for their properties and potential applications [51]. It is worth noting that controlling the particle size is crucial, as it can greatly influence the nanoparticles’ optical, electronic, and catalytic properties [63]. Furthermore, the spherical shape observed in the TEM images is a common characteristic of TiO_2_-NPs. Spherical nanoparticles often exhibit improved dispersibility, higher surface area, and enhanced stability, making them desirable for various applications such as photocatalysis, energy storage, and biomedical applications [64,65].

An EDX analysis was able to detect the elemental compositions of as-formed NPs. Figure 5C shows that TiO_2_ was successfully formed, as seen by absorption peaks with bending energies of 0.4 and 4.5 KeV, which belong to Ti ions, and 0.5 KeV, which corresponds to O ions. According to the EDX analysis, the main component of the sample was Ti, with weight and atomic percentages of 41.66 and 37.75%, respectively, followed by an O ion, with weight and atomic percentages of 31.76 and 35.6%, respectively. The EDX analysis also revealed the presence of C and Cl in the as-formed sample, with weight percentages of 25.21 and 1.37%, respectively (Figure 5C). The presence of C and Cl could be related to the scattering of a capping agent coming from the CFF of probiotic bacteria [66]. 

### 3.5. In Vitro Cytotoxicity of TiO_2_-NPs 

The goal of this experiment is to detect a safe dose of TiO_2_-NPs to be utilized for the biocontrol of fungal strain AL1 (the highest enzyme producers among fungi isolated from damaged leather binding). To achieve this goal, the in vitro cytotoxicity of probiotic-mediated biosynthesis of TiO_2_-NPs was evaluated against two normal cell lines, HFB4 and Wi38, using the MTT assay method [20] (Figure 6). The selection of these two normal cell lines is because the workers or visitors to libraries, museums, and archives handle books or historical manuscripts by hand (direct skin), and they are also inhaling dust and air at the site. Therefore, the selection of normal skin and lung cells is an important parameter. Data analysis showed that the activity of TiO_2_-NPs toward two normal cell lines was in a concentration-dependent manner. This finding was compatible with published studies about the cytotoxicity and biocompatibility of green synthesized NPs [67,68]. At a high concentration of 1000 µg mL^−1^, the cell viability percentages were 3.2 ± 0.8% for WI38 and 10.6 ± 1.6% for HFB4. These low-viability percentages indicate that the TiO_2_-NPs at this concentration have a significant negative impact on cell viability for both cell lines. However, as the concentration of TiO_2_-NPs decreases, the cell viability percentages increase. At a concentration of 125 µg mL^−1^, the viability percentages rise to 46.8 ± 1.0% for WI38 and 99.6 ± 0.4% for HFB4. This suggests that at this lower concentration, TiO_2_-NPs have a less detrimental effect on cell viability, particularly for HFB4, where almost all cells survive. Further reducing the concentration to 62.5 µg mL^−1^ resulted in a significant increase in cell viability for both cell lines, with the percentages reaching 99.9 ± 0.3% and 99.8 ± 0.9 for WI38 and HFB4, respectively. This indicates that at this concentration, TiO_2_-NPs have minimal or no adverse effects on the viability of these normal cells, as almost all cells survive. 

The mechanisms underlying the effects of TiO_2_-NPs on these cell lines can be complex and multifaceted. Several factors may contribute to the observed concentration-dependent cytotoxicity. (1) Dose-dependent cellular uptake: At higher concentrations, cells may take up larger amounts of TiO_2_-NPs, leading to increased cytotoxicity [69]. The nanoparticles may enter the cells through various mechanisms, such as endocytosis, and interfere with normal cellular processes. (2) Reactive oxygen species (ROS) generation: TiO_2_-NPs can generate ROS when exposed to light or undergo photocatalytic reactions. Higher concentrations may result in increased ROS production, leading to oxidative stress and cellular damage [65]. This could explain the lower cell viability of two cell lines at higher concentrations. (3) Surface properties and agglomeration: TiO_2_-NPs’ surface characteristics, such as size, charge, and agglomeration state, can influence their interactions with cells. Agglomerated nanoparticles may have different cellular interactions compared to individual nanoparticles. The concentration-dependent behavior could be related to changes in nanoparticle agglomeration and subsequent cellular uptake [70,71]. (4) Cell-specific responses: Different cell lines may exhibit varying sensitivities to TiO_2_-NPs due to differences in their inherent biological characteristics and signaling pathways [72,73]. The observed variations in cell viability between Wi38 and HFB4 could be attributed to variations in their response mechanisms and molecular profiles.

The IC_50_ (half maximal inhibitory concentration) values indicate the concentration of TiO_2_-NPs required to inhibit cell viability by 50% in the respective cell lines Wi38 and HFB4. The IC_50_ for Wi38 is 114.1 ± 8.1 µg mL^−1^, while the IC_50_ for HFB4 is 237.5 ± 3.5 µg mL^−1^. These values suggest that Wi38 cells are more sensitive to the cytotoxic effects of TiO_2_-NPs compared to HFB4 cells. Wi38 cells require a lower concentration of TiO_2_-NPs to reach 50% inhibition of cell viability compared to HFB4 cells. Based on these findings, using a concentration of 100 µg mL^−1^ of TiO_2_-NPs in leather (experimental study) can be considered safe for normal cells.

### 3.6. Experimental Study

#### 3.6.1. Assessment of Fungal Growth

*Penicillium expansum* AL1 was inoculated on leather samples treated with TiO_2_-NPs at a concentration of 100 µg mL^−1^. The results of the data analysis indicate that after 7 days of treatment with green synthesized TiO_2_-NPs, the growth inhibition percentage of *P. expansum* was found to be 73.2 ± 2.5%. This suggests that the NPs effectively inhibited the growth of the fungus on the leather surface. After 14 and 21 days, the inhibition percentages increased to 84.2 ± 1.8% and 88.8 ± 0.6%. This indicates a further improvement in the effectiveness of the NPs in inhibiting the growth of fungal strain AL1 over time (Figure 7). Although there is variation in the growth inhibition percentages over time, the analysis of variance (ANOVA) revealed that there was a significant difference between the inhibition percentages observed after 7, 14, and 21 days. This suggests that the NPs reached their maximum inhibitory effect within the first 21 days.

There are several possible mechanisms through which green synthesized TiO_2_-NPs could inhibit the growth of *P. expansum* on leather. For instance, when the NPs come into contact with the fungal cells, they may disrupt the cell membrane integrity, leading to the leakage of cellular components and ultimately causing cell death. This direct interaction between the NPs and the fungal cells could be one mechanism contributing to the inhibition of *P. expansum* growth [74]. Also, TiO_2_-NPs can generate reactive oxygen species (ROS), such as superoxide radicals and hydroxyl radicals. These ROS are highly reactive and can cause oxidative stress in the fungal cells. Oxidative stress can lead to damage to cellular structures and biomolecules, inhibiting the growth of *P. expansum* [75]. The presence of TiO_2_-NPs on the leather surface may create a physical barrier that hinders the growth and colonization of *P. expansum*. The nanoparticles can form a coating or adhere to the leather, preventing the fungus from accessing nutrients and establishing a favorable environment for growth [76]. Green synthesized TiO_2_-NPs can release metal ions when in contact with the fungal cells. These metal ions have known antimicrobial properties and can exert toxic effects on the fungal cells, leading to growth inhibition [77]. 

#### 3.6.2. Investigation of the Surface Morphology by Environmental Scanning Electron Microscope (ESEM) 

The results of SEM of the untreated and treated leather samples with and without inoculation after 21 days of aging are shown in Figure 8. As shown, the surface morphology of the untreated leather without fungal inoculation (negative control) was smooth, and the grouping of fine follicles was easily recognized (Figure 8A). In contrast, the spores of *P. expansum* were colonized on infected untreated leather causing deformation of the morphological form of collagen fibers (Figure 8B). Moreover, the surface was coarse and non-homogeneous, and the grain surface pattern cannot be recognized. The appearance of fungal hyphae was observed, and the spores cover the surface of all degraded samples. In Figure 8C, it is evident that the leather samples treated with TiO_2_-NPs and not inoculated with fungal strain AL1 were not affected by loaded NPs. This is observed by the collagen fibers maintaining contact in an organized and homogeneous network without any signs of deterioration. These results indicate that the TiO_2_-NPs did not adversely affect the components of the leather samples compared to the untreated samples. Thus, the TiO_2_-NPs appear to be a promising treatment for preserving the integrity of leather without compromising its structural components. 

On the other hand, leather samples treated with TiO_2_-NPs with the inoculated fungal strain AL1 showed a reduction in fungal growth (Figure 8D) compared to the infected untreated sample. It can be added that the use of TiO_2_-NPs protected the surface morphology from fungal growth. This is one of the most important requirements for the long-term preservation of cultural heritage in museums and libraries. 

#### 3.6.3. Attenuated Total Reflectance Fourier-Transform Infrared (FTIR-ART)

FTIR-ART analysis of the untreated leather sample (negative control) showed the characteristic collagen bands in the region between 3294 cm^−1^ and 2921 cm^−1^ (Figure 9). The mentioned bands are referred to amide A and amide B, which are associated with the stretching of peptide N-H groups involved in inter-chain hydrogen bonding. The FTIR spectra in the characteristic bands of leather formation in the negative sample were not significantly different from the samples treated with TiO_2_-NPs in the presence or absence of fungal inoculation, which indicates no effect of TiO_2_-NPs on the chemical composition of the treated samples. The amide I band at 1631 cm-^1^ for the leather sample treated with TiO-NPs in the absence of fungal inoculation appeared close to the amide I band in the negative control at 1633 cm-^1^. The amide I at the band mentioned above is primarily caused by the stretching vibrations of the peptide carbonyl group (C=O) coupled weakly with C-N stretching and N-H bending [39]. It is sensitive to local order, and its exact position is determined by the primary conformation and the hydrogen bonding pattern within the protein molecule. The amide II bands at 1546–1543 cm^−1^ for the leather treated with TiO_2_-NPs with/without microbial inoculation were associated with N-H bending and C-N stretching vibration. The amide III bands at 1237 cm^−1^ and 1238 cm^−1^ for the leather treated with TiO_2_-NPs with/without microbial inoculation were associated with N-H in-plane bending and CH_2_ wagging vibration of the glycine backbone and proline side chain. The results proved that the collagen bands of leather inoculated with fungal strain AL1 without TiO_2_-NP treatment (positive control) shifted to a lower value compared to the leather without inoculation that was untreated (negative control). It was clear the intensities of bands of the infected sample (positive control) were lower than the intensities of the untreated sample before fungal infection (negative control), and this indicated that some changes occurred in the chemical composition of leather samples as a result of the fungal impact. The same observation was recorded for the intensities of the bands of the infected treated sample with TiO_2_-NPs, which were lower than the treated sample before the fungal infection. It was also noticed that the changes in the chemical composition of the infected treated sample were too low compared to the changes obtained from the infected or untreated sample. This indicates the efficiency of TiO_2_-NPs used for the treatment to protect the leather from fungal impact.

#### 3.6.4. Measurement of Color Change

The results (Table 2) of the color change in the untreated samples before and after infections with fungus and the treated samples before and after inoculation with fungus are as follows. L* value: the lightness (L* value) for the positive control (inoculated with fungal strain AL1) showed high change with the value of 61.2, 56.4 and 49.2 after 7, 14 and 21 days of aging, respectively, compared to 77.8 of the negative control. The results of the L* value for the leather treated with TiO_2_-NPs with/without fungal inoculation revealed a slight difference in lightness, which was very close to the L* value of the negative control.

The results of a* value of all samples studied are in red. The a* of the positive control showed an increase of 26.2, 29.1, and 33.1 after 7, 14, and 21 days of aging, respectively, compared to 20.1 in the negative control. A non-significant change in a* value was recorded in the treated leather samples by TiO_2_-NPs with/without fungal inoculation compared to the negative control. 

The results of the b* value tend to be in yellow. The b* value of the positive control showed an increase of 22.3, 26.5, and 29.3 after 7, 14, and 21 days of aging, respectively, compared to 18.1 in the negative control. On the other hand, there was a slight variation in the b* value between the negative control and the leather treated by TiO_2_-NPs with/without fungal inoculation. 

Total color change (ΔE): the results showed that there is a high change in the total color difference of the positive control sample, which was 18.1, 24.6, and 33.3 after 7, 14, and 21 days of aging, respectively, compared to the negative control sample. 

The results showed that the total color difference of the treated samples by TiO_2_-NPs with/without microbial inoculation was too low compared to the positive control. ΔE of the treated sample before infection was between 0.93 and 2.9, which cannot be recognized by the critical eye. ΔE of the infected treated samples was between 4.7 and 8.4, which was lower than the positive control sample. The results obtained proved that the treatment with TiO_2_-NPs gave high stability against fungal deterioration. 

Matsuo et al. [78] proved that the color changes during natural aging could be mainly explained as mild thermal oxidation. Moreover, the heat aging process changed the color values and total color values of the historical artifacts, as reported previously [79]. The aging process leads to the darkness of a color, which increased with increasing the aging time. The color changes could be due to the production of pigments and organic acids by fungi. Moreover, the microbial attack also causes multi-colored spots and white films [12]. 

#### 3.6.5. Mechanical Properties Measurement

The results obtained (Table 3) showed the results of tensile strength and elongation of all the studied samples. These results proved that there was a high reduction in tensile strength in the positive control. The reduction percentage was 21%, 37%, and 44% after 7, 14, and 21 days of fungal infection compared to the negative control sample. The results of the treated samples with TiO_2_-NPs led to an improvement in tensile strength since the reduction in this parameter was not significant. The reduction in the treated samples was 2%, 3%, and 4% after 7, 14, and 21 days. The reduction in tensile strength of the infected treated samples was 9%, 12%, and 14% after 7, 14, and 21 days of fungal infection compared to the negative control sample. For elongation, the reduction in the positive control sample was 25%, 33%, and 37% after 7, 14, and 21 days, respectively, compared to the negative control sample. The reduction in the treated samples with TiO_2_-NPs before fungal infection was 16%, 25%, and 25%. After fungal infection, the reduction in the treated samples was 18%, 29%, and 29% after 7, 14, and 21 days, respectively. All the results of tensile strength and elongation of the treated before and after fungal infection proved that the treatment of leather samples with TiO_2_-NPs improved this parameter. These results were confirmed by Hassan et al. [2], who proved that the use of nanomaterials improves the mechanical properties of materials.

## 4. Conclusions

The historical manuscript studied exhibited severe deterioration, likely due to inappropriate storage conditions and high humidity before its arrival at its current place in the basement of the Library of Alexandria. Various deterioration aspects were observed, including hardness, fragility, inflexibility, yellowing of paper, and stains. Fourteen fungal strains were isolated, with four from the leather binding and eleven from the paper sheets. The identified fungi included *Penicillium expansum*, *Eurotium* spp., *Aspergillus nidulans*, *A. flavus*, *A. niger*, and *Paecilomyces* spp., and *Penicillium citrinum. P. chrysogenum,* and *A. niger* exhibited the highest cellulase activity, while *P. expansum* showed the highest amylase and gelatinase activities. TiO_2_-NPs were synthesized by an eco-friendly approach using a probiotic bacterial strain, *L. plantarum,* and characterized using UV-Vis spectroscopy, FT-IR analysis, X-ray diffraction, and TEM. The in vitro cytotoxicity test confirmed the safety of TiO_2_-NPs (at a concentration of 100 µg mL^−1^) to use as a treatment for leather to inhibit fungal growth. Treatment with TiO_2_-NPs inhibited *P. expansum* on leather with percentages of 73.2 ± 2.5%, 84.2 ± 1.8%, and 88.8 ± 0.6% after 7, 14, and 21 days, respectively. SEM and FTIR-ART analyses revealed improved surface morphology and chemical composition compared to the positive control sample (leather inoculated with fungal strain AL1 in the absence of NPs). Additionally, TiO_2_-NPs improved mechanical properties and reduced color differences. Based on these findings, TiO_2_-NPs fabricated by the green method are recommended for the treatment of leather bindings to inhibit fungi and hence reduce biodeterioration.

## Figures and Tables

**Figure 1 biology-12-01025-f001:**
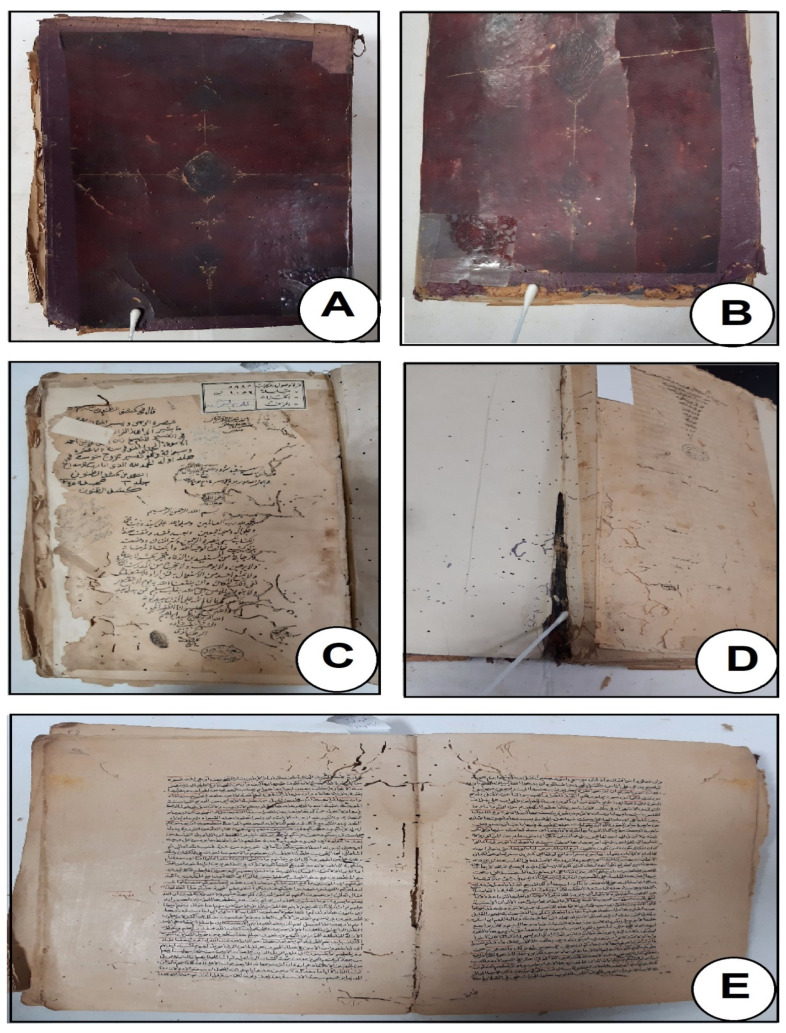
Some deterioration aspects on the papers and leather binding of a historical manuscript studied and deposited at Bibliotheca Alexandrina Library, Egypt. (**A**,**B**) are the cover and bookbinding leather, (**C**–**E**) are the historical paper sheet showing the different deterioration signs.

**Figure 2 biology-12-01025-f002:**
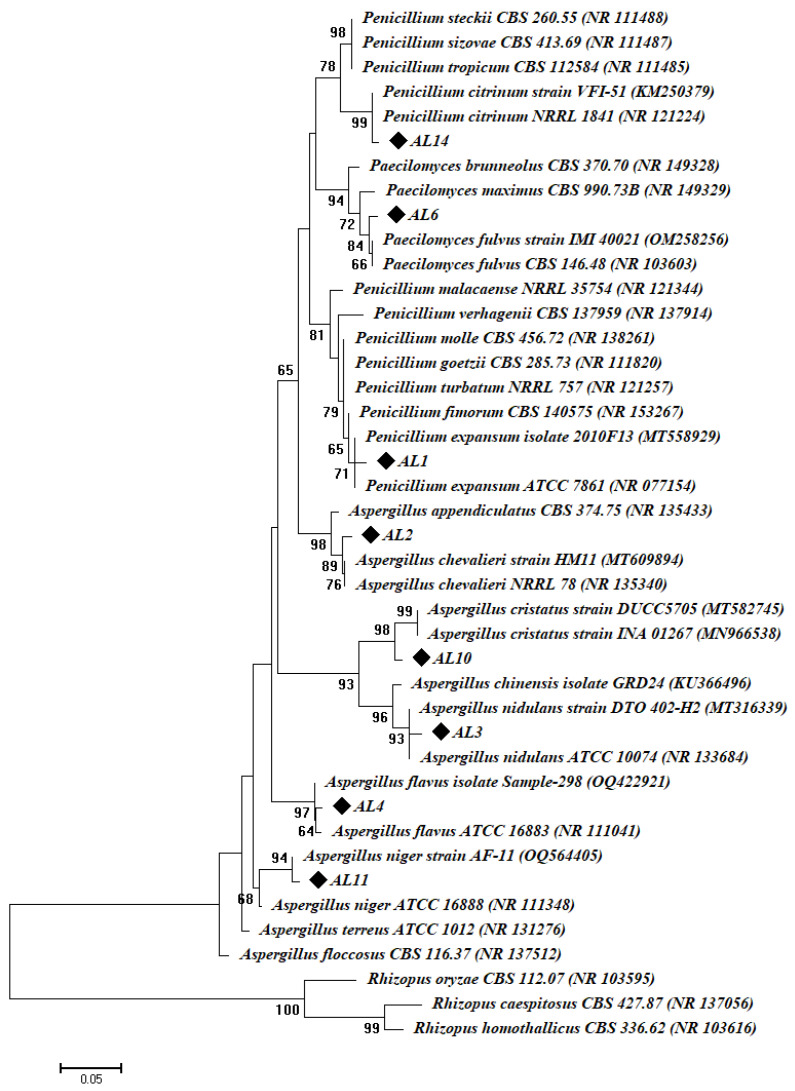
Phylogenetic tree of the selected fungal strains isolated from the deteriorated manuscript. This tree was constructed using the neighbor joining method with a bootstrap value of 1000 replicates. ◆ refer to the obtained fungal isolates subjected to ITS sequence analysis.

**Figure 3 biology-12-01025-f003:**
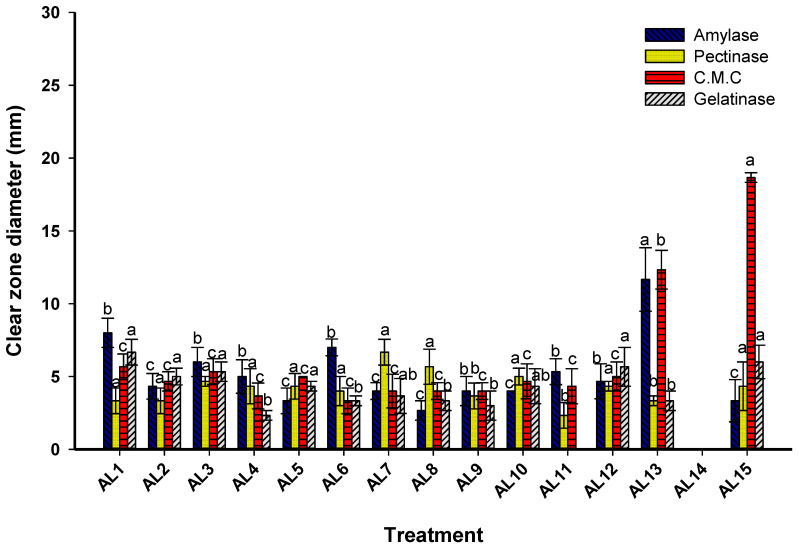
The enzymatic activity of different fungal strains isolated from a historical manuscript (paper and leather bookbinding). Different letters between various fungal strains on bars for the same enzyme indicate that the mean values of the clear zone are significantly different (*p* ≤ 0.05) (n = 3).

**Figure 4 biology-12-01025-f004:**
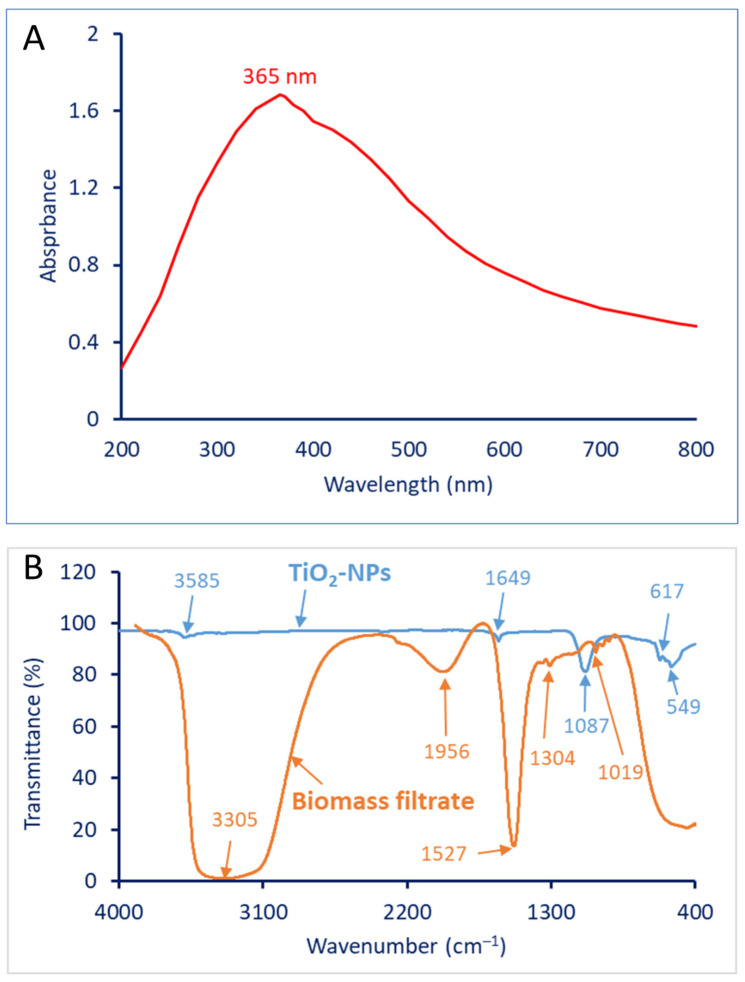
Characterization of TiO_2_-NPs fabricated by the CFF of probiotic bacteria, *L. plantarum*. (**A**) UV-Vis spectroscopy showing the maximum SPR peak at 365 nm, (**B**) FT-IR chart for the CFF and TiO_2_-NPs.

**Figure 5 biology-12-01025-f005:**
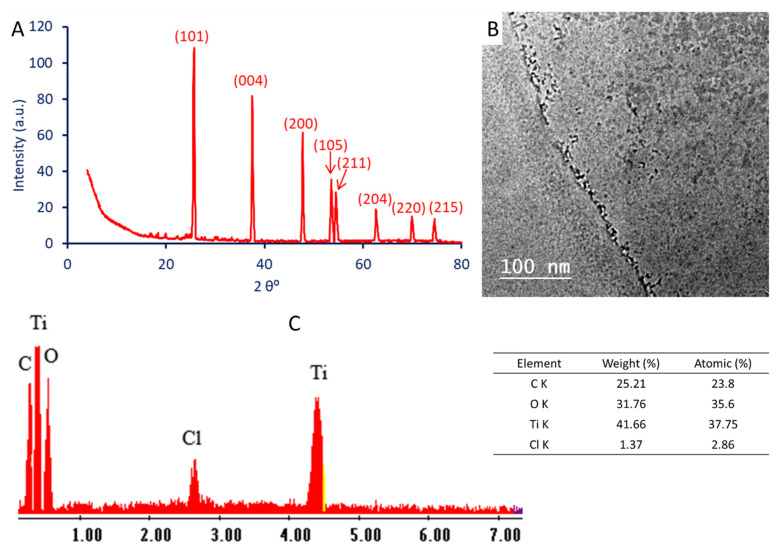
(**A**) The X-ray diffraction pattern of TiO_2_-NPs synthesized by probiotic *L. plantarum*, (**B**) TEM analysis showing a spherical shape, and (**C**) EDX analysis showing elemental mapping of TiO_2_-NPs.

**Figure 6 biology-12-01025-f006:**
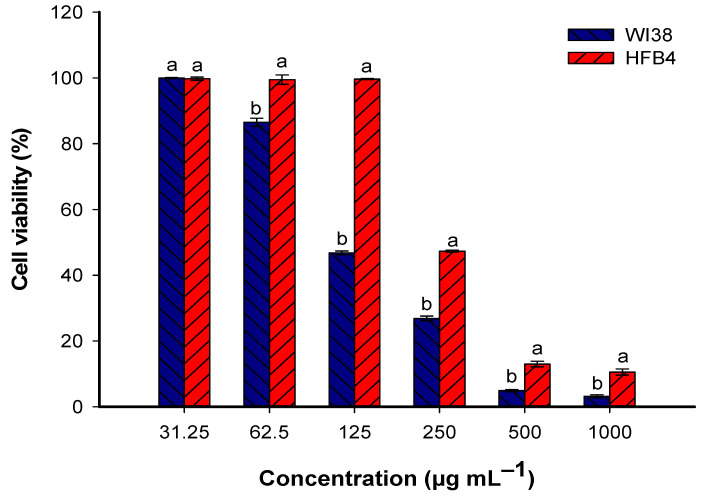
In vitro cytotoxicity of TiO_2_-NPs against two normal cell lines, WI38 and HFB4, after treatment with different concentrations. Different letters on bars for the same concentration indicates that the mean value of cell viability is significantly different (*p* ≤ 0.05) (n = 3).

**Figure 7 biology-12-01025-f007:**
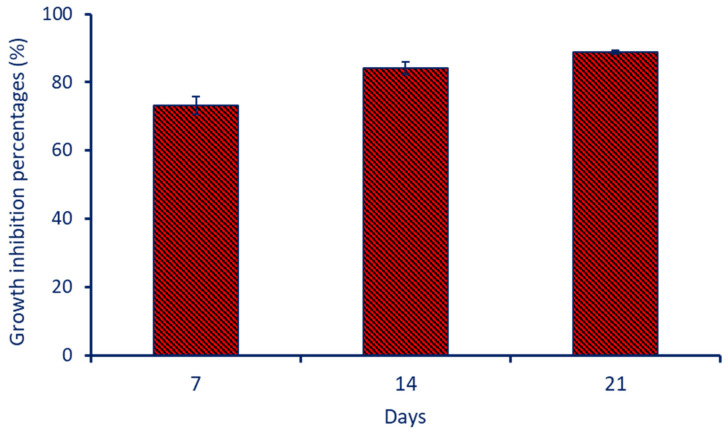
Growth inhibition percentages of *P. expansum* AL1 grown on leather treated with bacterial synthesized TiO_2_-NPs at a concentration of 100 µg mL^−1^ after incubation periods of 7, 14, and 21 days.

**Figure 8 biology-12-01025-f008:**
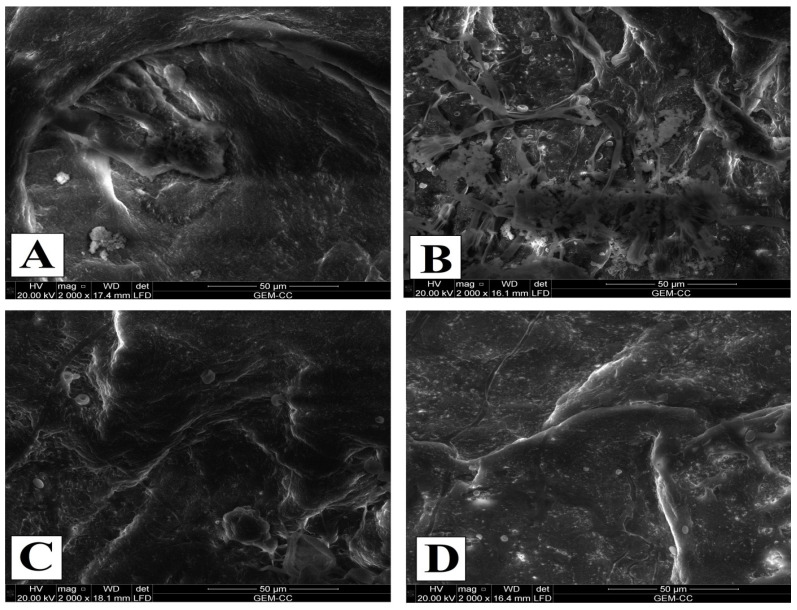
ESEM for leather samples before and after TiO_2_-NP treatment in the presence and absence of fungal inoculation. (**A**) The negative control (leather sample without NP treatment and in the absence of fungal inoculation), (**B**) the positive control (leather sample without TiO_2_-NP treatment and inoculated with fungal strain), (**C**) the leather sample treated with TiO_2_-NPs in the absence of fungal inoculation, and (**D**) the leather sample treated with TiO_2_-NPs and inoculated with *P. expansum*.

**Figure 9 biology-12-01025-f009:**
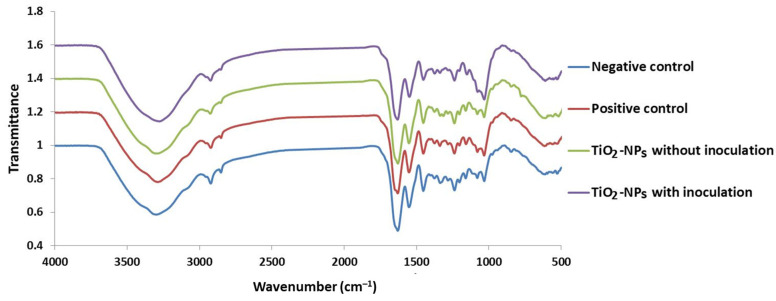
FTIR-ART chart for the treated/untreated leather samples with TiO_2_-NPs in the presence and absence of fungal inoculation.

**Table 1 biology-12-01025-t001:** Summary of the traditional and molecular identification of different fungal strains isolated from the deteriorated historical manuscript.

Fungal Code	Source of Isolation	Traditional Identification	Molecular Identification	GenBank Accession Number	Closest Accession Number	Similarity Percentages
AL1	Leather	*P. expansum*	*P. expansum*	OR052528	MT558929	99.14
AL2	Paper	*E. chevalieri*	*A chevalieri*	OR052529	MT609894	99.56
AL3	Leather	*A. nidulans*	*A. nidulans*	OR052530	MT316339	99.48
AL4	Paper	*A. flavus*	*A. flavus*	OR052531	OQ422921	98.78
AL5	Leather	*A. flavus*	–	–	–	–
AL6	Paper	*Paecilomyces* sp.	*Paecilomyces fulvus*	OR052532	OM258256	99.24
AL7	Paper	*Paecilomyces* sp.	–	–	–	–
AL8	Paper	*A. nidulans*	–	–	–	–
AL9	Paper	*A. flavus*	–	–	–	–
AL10	Paper	*E. herbariorum*	*A. cristatus*	OR052533	MT582745	99.03
AL11	Paper	*A. niger*	*A. niger*	OR052534	OQ564405	99.06
AL12	Paper	*A. flavus*	–	–	–	–
AL13	Leather	*P. expansum*	–	–	–	–
AL14	Paper	*P. citrinum*	*P. citrinum*	OR052535	KM250379	97.98
AL15	Paper	*P. citrinum*	–	–	–	–

– means that this isolate was identified by the traditional method only.

**Table 2 biology-12-01025-t002:** Color change measurement of the untreated/treated leather samples by TiO_2_-NPs with/without inoculation of *P. expansum* AL1.

Leather Treatment	After 7 Days				After 14 Days				After 21 Days			
	L*	a*	b*	(ΔE)	L*	a*	b*	(ΔE)	L*	a*	b*	(ΔE)
A	77.8	20.1	18.1	0.00	77.8	20.1	18.1	0.00	77.8	20.1	18.1	0.00
B	61.2	26.2	22.3	18.1	56.4	29.1	26.5	24.6	49.2	33.1	29.3	33.3
C	77.3	20.6	18.8	0.93	76.0	20.9	18.9	2.1	75.8	21.6	19.7	2.9
D	73.8	21.9	19.8	4.7	72.4	22.6	20.4	6.2	70.8	23.3	21.7	8.4

A is the negative control (leather sample without TiO_2_-NPs in the absence of fungal inoculation); B is the positive control (leather sample without treatment and inoculated with fungal strain AL1); C is the leather treated with TiO_2_-NPs without fungal inoculation; and D is the leather treated with TiO_2_-NPs with fungal inoculation.

**Table 3 biology-12-01025-t003:** Tensile strength (N/mm^2^) and elongation (%) for the untreated/treated leather samples with TiO_2_-NPs before and after fungal inoculation.

Leather Treatment	After 7 Days		After 14 Days		After 21 Days	
	Tensile Strength (N)	Elongation %	Tensile Strength (N)	Elongation %	Tensile Strength (N)	Elongation %
A	440.5	27.80	438.5	26.20	436.2	24.10
B	346.8	20.90	274.8	17.48	245.2	16.40
C	430.1	23.28	425.1	19.65	420.9	18.00
D	400.20	22.70	385.4	18.55	375.6	17.09

A is the negative control (leather sample without TiO_2_-NPs and in the absence of fungal inoculation); B is the positive control (leather sample without treatment and inoculated with fungal strain AL1); C is the leather treated with TiO_2_-NPs without fungal inoculation; and D is the leather treated with TiO_2_-NPs with fungal inoculation.

## Data Availability

The data presented in this study are available upon request from the corresponding author.

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
