# Peer review of "An Eco-Friendly Approach Utilizing Green Synthesized Titanium Dioxide Nanoparticles for Leather Conservation against a Fungal Strain, Penicillium expansum AL1, Involved in the Biodeterioration of a Historical Manuscript"

_biology, 2023, doi:10.3390/biology12071025_

Round 1

Reviewer 1 Report

The paper "Eco-Friendly Approach Utilizing Green Synthesized Titanium Dioxide Nanoparticles for Leather Conservation Against Fungal Strain, Penicillium expansum AL1, Involved in Biodeterioration of a Historical Manuscript" is certainly interesting for the use of titanium oxide nanoparticles (TiO2-NPs) of green synthesis to be applied to historical manuscripts deteriorated biologically.

Many analyses are presented that reveal the potential of the application of TiO2-NPs in the inhibition of fungal attack on manuscripts and then on the leather and paper.

My perplexity, however, is about applying the treatment directly to the manuscript because in section 2.7 the sample used for testing, is immersed for 2 minutes in a solution containing TiO2-NPs, so I wonder if you wanted to do it practically on a historical manuscript, wouldn't you risk ruining it even more?

Have you thought about applying the nanoparticles differently, without immersion, perhaps with an inert carrier?

Another doubt concerns the use of the two cell lines W138 and HFB4. First, in the Abstract and Introduction it talks about cytotoxicity, while in Materials and Methods it talks about biocompatibility. These seem like two different terms, can you please explain further.

Furthermore, could you please explain the rationale behind the specific selection of these two strains?

The work, while presenting remarkable results, is a bit too long. There are sentences that are really too short and some repeated. Some information returns throughout the text. In discussions, one sometimes loses the thread because there are too many numbers, which are all referenced in the tables anyway. My advice is to lighten the content.

The Conclusions are a summary of what has gone before, but especially the last sentence leaves the work unfinished because the tests were only done in vitro.

Specific comments:

Line 136: Why is cleaned term used?

Line 142: also write the PDA’s extended name in uppercase.

Lines 149-160: Molecular analyses are explained too briefly and there is no bibliographic reference for the primers used.

Line 155; 1 mL instead of 1 L

Line 156: 50 mL instead of 50 L

Line 164: Write in parenthesis what is meant by the MSA medium.

Line 181: MRS as above

Lines 190-191 and Lines 195-196 is repeated the same sentence on the color of the synthesis of TiO2-NPs, choose where to write it.

Line 224: 2.5.4. Biocompatibility of TiO2-NPs or Cytotoxicity? Lacking bibliography for cell line selection

Line 234: medium RPIM, please specify a brand.

The TiO2-NPs concentration was doubled (1000 - 31.25 μg mL-1). As written, it is not clear how many concentrations were used; you have to look up the figure to figure it out.

Lines 254-255: by the authors according to the standard. Which ones? Is there a reference bibliography?

Line 258: 2.7. Experimental design. It is not very clear how the inoculation of the fungus takes place, explain how the three-discs (5 mm) are made.

Line 273: 2.7.1. Assessment of fungal growth. I think perhaps an explanatory photo of the measurements would have fit here.

Line 320: Figure 1 not bold

Line 323: Vichi et al. not bold

Line 353: 3.2. Fungal isolation and identification. Paragraph too long, refer more to Table 1 where everything is written, it is useless to report for example all accession numbers.

Line 433: These enzymes were the focus of the current investigation: I think this sentence should be changed because it seems that this is only the focus of the work.

Figure 3: since there are so many results to show, I would avoid the error bars which also do not read well.

Lines 478/485/487: Correct Lactobacillus Plantarum with Lactobacillus plantarum

Lines 480-483: I think it would be more correct to fix the sentences between 3.4 and 3.4.1

Figure 4: Correct Lactobacillus Plantarum with Lactobacillus plantarum

Lines 501/ 504: Correct Lactobacillus Plantarum with Lactobacillus plantarum

Line 655: Fig. 8a A capitalized.

Lines 661-666: It is not clear why if “in the sample with TiO2-NPs and without inoculated fungal strain” the non-presence of fungal spores is discussed. Please explain better.

Lines 679-681: twice in the same sentence is written Figure 9, remove one.

Figure 9: ATR: not bold

Line 754: Table 3 not bold 

The English language is quite understandable, the biggest problem is the short and repeated sentences in the text.

Author Response

Dear reviewer, thank you very much for your comments. All comments were answered point-by-point as shown in the author's response file. "Please see the attachment."

Reviewer 2 Report

This paper deals with protection of the cultural heritage and therefore is very significant for the whole humankind. From my side there are some remarks:

- L52 - common is FTIR-ATR
- L59-61 - the sentence is unclear
- L65 - although authors state that TiO2 is nontoxic, there are some sources to state it is low-toxic (Ahmed, O.B.; Alamro, T. Evaluation of the Antibacterial Activities of Face Masks Coated with Titanium Dioxide Nanoparticles. Sci Rep 2022, 12, doi:10.1038/s41598-022-23615-w) and it is still under study (Franco-Castillo, I.; Hierro, L.; de la Fuente, J.M.; Seral-Ascaso, A.; Mitchell, S.G. Perspectives for Antimicrobial Nanomaterials in Cultural Heritage Conservation. Chem 2021, 7, pp. 629–669, doi:10.1016/j.chempr.2021.01.006)
- L115 - again, FTIR-ATR
- L140 - "aa" remove
- L159 - explain software mentioning
- L189 - suppose it should state 1M NaOH
- L206 - "-" should be in superscript
- L323 - why is the citation in Bold?
- L501, L502 - write the abbreviation of FTIR in same way
- L630 - on which proof do the authors provide information that NPs reached maximum inhibitory effect within first 21 day. The results in Fig 7 suggest that on 21st day they reached maximal value in investigated period, but there is no results provided that in , for example, 25 day it would not be even higher
- L720 - brightness is not the same as lightness
- Table 2 - please shorten the treatment name or make numbers central to the text to make it easier to read
- Conclusion should be more concise and stress just the significance of the research, not all the results
- L801 - the inhibition percentage is not the same as in results section

Quality of English is of satisfactory level.

Author Response

(The authors gave the same response as above.)
